# Memory and relatedness of transcriptional activity in mammalian cell lineages

Nicholas E. Phillips [1], Aleksandra Mandic[1], Saeed Omidi[1], Felix Naef [1] & David M. Suter [1]

Phenotypically identical mammalian cells often display considerable variability in transcript levels of individual genes. How transcriptional activity propagates in cell lineages, and how this varies across genes is poorly understood. Here we combine live-cell imaging of short-lived transcriptional reporters in mouse embryonic stem cells with mathematical modelling to quantify the propagation of transcriptional activity over time and across cell generations in phenotypically homogenous cells. In sister cells we find mean transcriptional activity to be strongly correlated and transcriptional dynamics tend to be synchronous; both features control how quickly transcriptional levels in sister cells diverge in a gene-specific manner. Moreover, mean transcriptional activity is transmitted from mother to daughter cells, leading to multi-generational transcriptional memory and causing inter-family heterogeneity in gene expression.

[1] Institute of Bioengineering, School of Life Sciences, Ecole Polytechnique Fédérale de Lausanne, CH-1015 Lausanne, Switzerland. These authors contributed equally: Nicholas E. Phillips, Aleksandra Mandic. Correspondence and requests for materials should be addressed to F.N. (email: felix.naef@epfl.ch) or to D.M.S. (email: david.suter@epfl.ch)

Major changes in transcriptional states that propagate through cell generations is characteristic of embryonic development. Such dynamics often result in irreversible changes in phenotypic states that are then transmitted through cell division[1]. In the Waddington's landscape representation of cell types, this is akin to transitions between distinct metastable states in gene expression space[2,3]. In addition to these genome-wide alterations of gene expression profiles associated with different cell types, even phenotypically identical cells display significant intercellular variability and temporal changes in the levels at which individual genes are expressed[4–6]. The temporal characteristics of these gene expression fluctuations can be interpreted as memory, in particular the time needed to observe significant changes in the levels of molecular species such as RNAs or proteins. For proteins, expression levels and fluctuations are controlled on multiple levels, including via the half-lives of gene expression products (e.g., proteins and mRNAs), but also through the time-scales of transcriptional fluctuations. When gene expression memory exceeds one cell generation, the levels of gene expression will be related within families of cells. Such trans-generational transcriptional memory might then prime downstream-spatial-gene expression patterns, for instance in solid tissues where cells sharing a common ancestor typically remain in close proximity.

In general, gene expression fluctuations can be caused by diverse sources, such as intrinsic noise resulting from the randomness in biochemical processes controlling gene expression, as well as extrinsic variability caused by differences in cellular parameters[7], such as size[8,9], mitochondrial content[10,11], cell cycle stage[8,12–14], differences in cellular microenvironment[11,15,16], or transitions between different phenotypic states[17,18]. Importantly, these diverse sources of variability are linked with distinct time scales. For example, transcriptional bursting causes intrinsic fluctuations with a time scale on the order of one to several hours[19–21], while extrinsic fluctuations in cellular parameters can be significantly longer-lived, and easily exceed one cell generation[22].

Several studies have investigated different aspects of gene expression memory on the protein level. For instance, in mouse embryonic stem cells (mESCs) exhibiting reversible phenotypic transitions between naïve and primed states, it was found that transitions between different NANOG protein levels can exceed one generation, and after sorting for low NANOG levels there is a subpopulation without NANOG onset for 70 h, presumably as a consequence of these transitions[18]. In H1299 lung carcinoma cells, the duration of gene expression memory was estimated directly at the protein level, and found to typically last between 1 to 3 cell cycles[23]. For proteins, such memory may largely reflect mRNA and protein half-lives[24], which often exceed the duration of the cell cycle[25]. Only few studies investigated the dynamics of transcriptional fluctuations and associated memory. For example, transcriptional parameters in *Dictyostelium* were found to be correlated both between sister and mother-daughter cells[26]. In the developing *Drosophila* embryo, higher transcriptional activity in mother nuclei increases the probability of rapid re-activation in daughter nuclei[27]. However, very little is known about the timescales of transcriptional memory in mammalian cells in lineages of phenotypically identical cells.

Here, we use short-lived transcriptional reporters to determine how transcriptional fluctuations are propagated over time and across cell division in phenotypically homogenous mESCs. We find that genes differ broadly in the dynamics of their transcriptional fluctuations at both short (in the hour range) and long (cell generations) time-scales, which results in large differences in the propagation of transcriptional activity. We also find a remarkably large correlation in transcriptional activity of sister cells, suggesting that inherited factors from the mother cell and/or similarity in cellular microenvironment contribute to transcriptional dynamics in dividing cells. Extending our analysis to pairs of mother-daughter cells shows that mean transcriptional activity is reliably transmitted across generations, and after two generations cells are clustered around family mean levels. Thus, the relatedness of transcriptional activity in sibling cells and its transmission to daughter cells both structure gene expression fluctuations across lineages of phenotypically homogenous cells.

## Results

**Signatures of transcriptional fluctuations are gene-specific**. To monitor how transcriptional levels fluctuate and propagate over cell generations, we inserted a short-lived transcriptional luminescent reporter by gene trapping into endogenous genes (Supplementary Fig. 1). This method allows sensitive monitoring of transcriptional activity by luminescence imaging at high-time resolution without observable toxicity over long periods of time[20]. In total, we produced eight different gene trap cell lines, and an additional cell line where a construct driving the expression of the short-lived luciferase from the pGK promoter was integrated as a single copy in the genome[20]. The insertion sites of the constructs were mapped using splinkerette PCR (Supplementary Fig. 2)[28]. To analyse how temporal transcriptional activity profiles compare both in pairs of mother–daughter, as well as sister cells (Fig. 1a, b), we monitored total transcriptional reporter levels with a time resolution of 5 min, and manually tracked approximately 50 pairs of sister cells per cell line from division to division to obtain single-cell traces. In addition, for three clones we quantified transcriptional activity profiles of mother and daughter cells over two cell generations.

We first aimed to determine whether differences in transcriptional levels across cells decayed quickly or if they were maintained over longer timescales and transmitted to daughter cells (Fig. 1c). The live-cell imaging of sister cells generated pairs of time traces, and exploratory data analysis revealed several key features of transcriptional dynamics. First, the mean and spread of transcriptional reporter levels across the population of cells in function of time were gene-specific (Fig. 2a, Supplementary Fig. 3). The average transcriptional reporter levels across the population increased during G1 phase (see Methods for cell cycle phase definition), consistent with RNA-seq analysis of pre-mRNA around the cell cycle[29], and then stayed approximately constant during S and G2 phases for most genes. Sorting cells by initial transcriptional reporter levels showed that for the *pGK* clone, cells tended to retain their relative expression levels for longer than for the *Dstn* gene (Fig. 2a). For *pGK*, transcriptional activity fluctuated around largely different mean levels in individual cells (Fig. 2b), suggesting that cells retained their average transcriptional levels over longer times than for *Dstn*. Unexpectedly, transcriptional profiles of sister cells often showed striking similarity over the cell cycle (Fig. 2c). Moreover, sister cells showed high correlation in reporter levels immediately following cell division, as explained from the partitioning of reporter protein and mRNA molecules. This sister-cell correlation then decreased over the cell cycle in a gene-specific manner at a slower pace than non-sister control pairs matched for similar initial levels (Fig. 2d, all genes shown in Supplementary Fig. 4), suggesting that transcriptional activity is transmitted along cell lineages.

**A hierarchical model for transcription in sister cells**. Next, we developed a mathematical model to quantitatively assess how transcriptional activity fluctuates in pairs of sister cells, taking into account the features described in Fig. 2. Previous studies

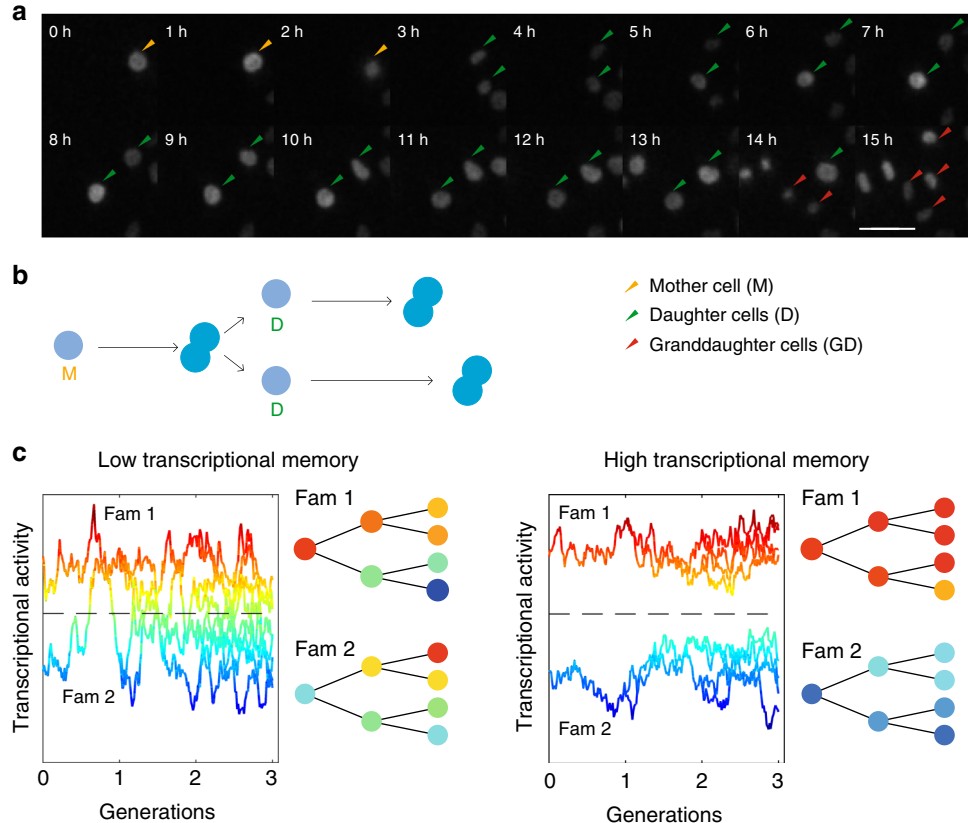

**Fig. 1** Monitoring the propagation of transcriptional activity in proliferating cells. **a** A cell from the *Rbpj* reporter line progressing through two cell cycles. Luminescent cell nuclei are tracked manually. Scale bar: 10 μm. **b** Representation of events in **a**. **c** Schematic of the transcriptional activity profiles over cell lineages in genes with short or long memory. Two families with either high or low initial transcriptional activity with respect to the population average (dotted black line) are shown

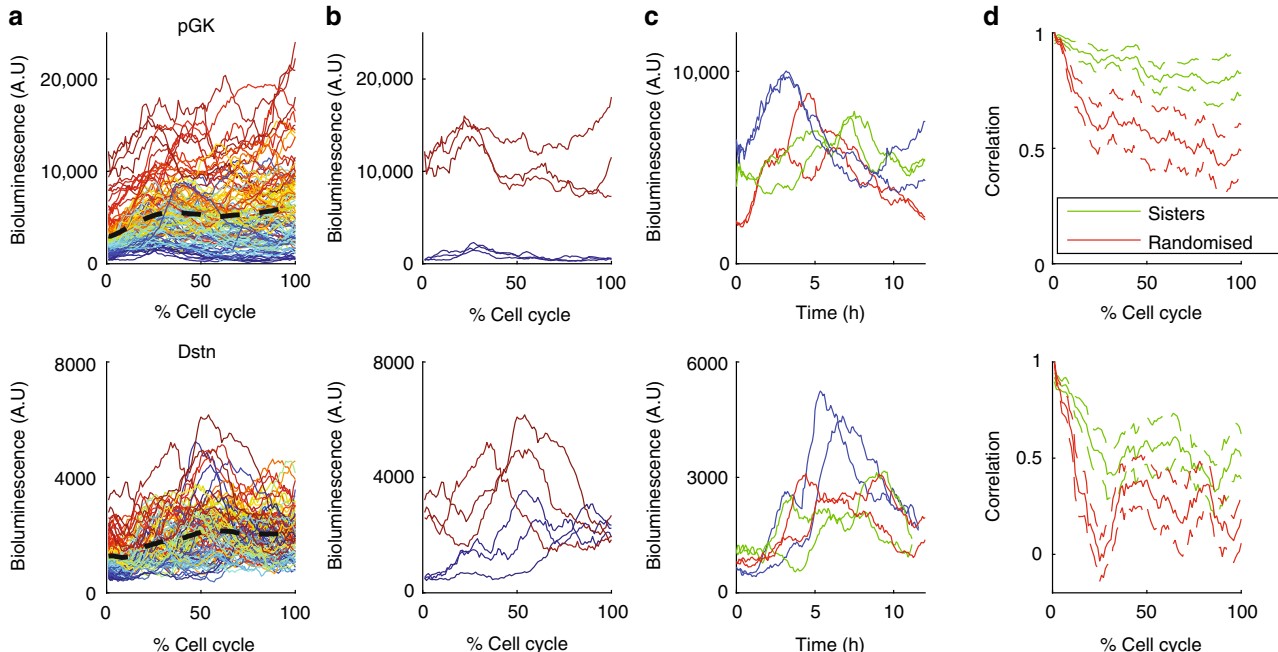

**Fig. 2** Single-cell reporter profiles and their correlation between sister cells. **a** Single-cell transcriptional reporter time series (total intensity per cell) for two genes (top: *Pgk*, 59 pairs of cells; bottom: *Dstn*, 50 pairs), measured from one cell division to the next (time is expressed in % of cell cycle time). Cells are colour-coded according to the ranking of the initial reporter level within the population. Dotted black line: population mean. **b** The top three cells with the highest/lowest initial reporter levels. **c** Examples of three pairs of sister cells (sister cells have the same colour). **d** The decrease in correlation between sister cells over the cell cycle. Green: correlation between sister cells; red: correlation between random cells, where each cell is matched with a non-sister with the nearest initial values. Error bars denote standard deviations obtained with bootstrap sampling

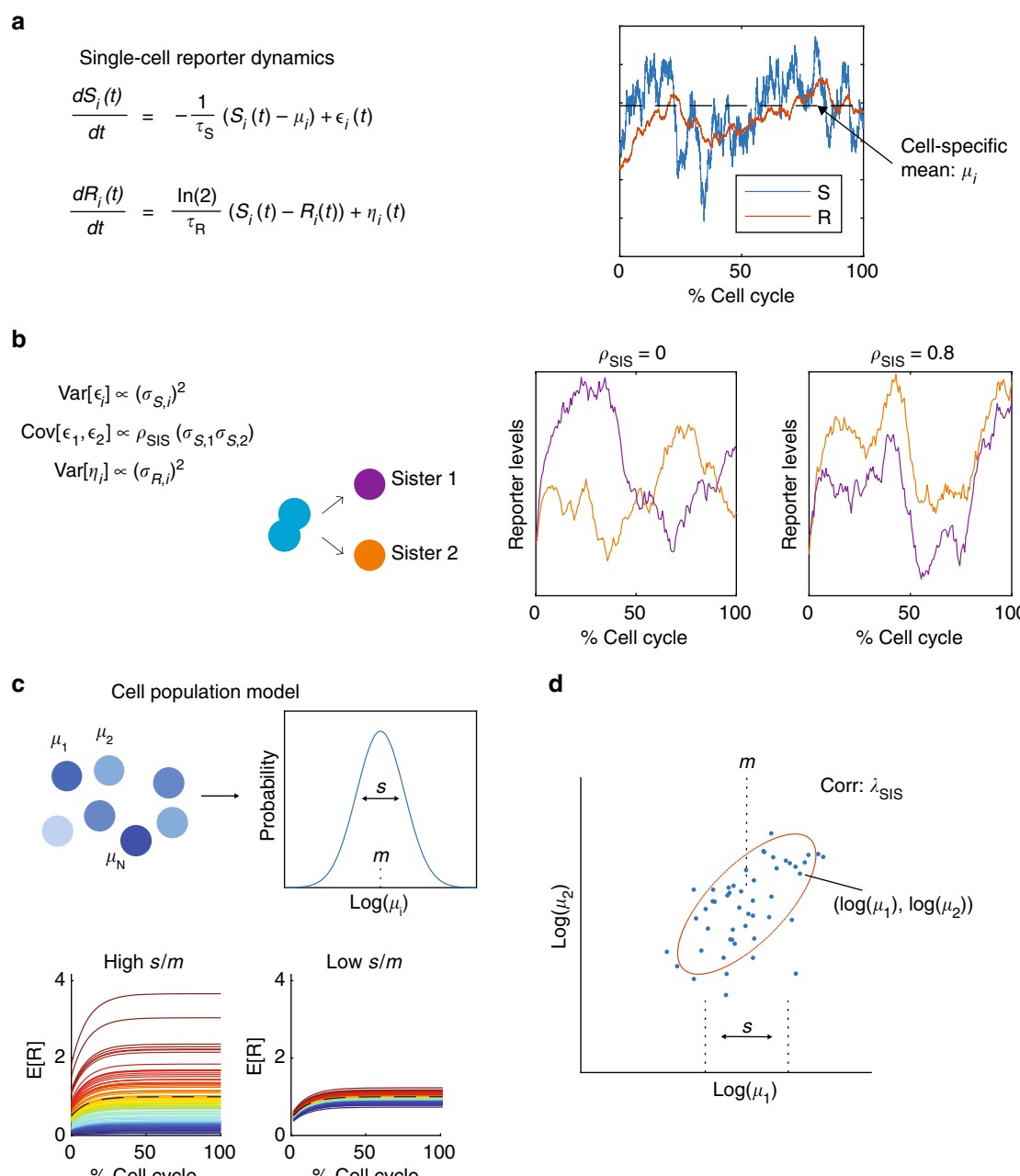

**Fig. 3** A hierarchical model of single-cell reporter dynamics in pairs of sister cells. **a** Single-cell dynamics are modelled probabilistically using stochastic differential equations (Methods). Each cell has a transcriptional activity ($S$) and a bioluminescent reporter ($R$) variable, where $S$ controls the production of $R$. **b** To account for stochastic fluctuations, both $S$ and $R$ are perturbed by noise terms $\varepsilon$ and $\eta$, respectively. The transcriptional noise ($\epsilon$) experienced in the two sister cells is correlated with parameter $\rho_{SIS}$, which describes whether sister cell dynamics are independent ($\rho_{SIS} = 0$) or if they share a similar shape over the cell cycle ($\rho_{SIS} > 0$). **c** The mean level of $S$ is cell-specific and denoted by $\mu_i$ for cell $i$, and the strength of the noise terms for $S$ and $R$ are also cell-specific and are denoted by $\sigma_{S,i}$ and $\sigma_{R,i}$, respectively. The distribution of cell-specific parameters $\mu_i$, $\sigma_{S,i}$ and $\sigma_{R,i}$ are described at the population level with log-normal distributions. $s$ describes the population level variability in cell-specific means. **d** The correlation of mean transcriptional levels between sister cells is quantified with $\lambda_{SIS}$

modelled the levels of gene expression products using birth-death processes, typically at steady-state and without modelling extrinsic fluctuations[20], with some exceptions[30]. Here, we use a simplified description using continuous variables, which retains essential properties and time scales of birth-death processes underlying gene expression, such as transcriptional bursting and reporter half-lives. The model can at the same time flexibly cope with non-steady state fluctuating transcription, fluctuating external parameters, and also be sufficiently tractable to allow efficient inference on the whole population level (Fig. 3).

Specifically, the model describes fluctuations in transcriptional reporter data at both the single-cell and population level (i.e., the set of all time traces for paired sister cells). In addition, this framework can also readily be applied to lineage related cells to quantify differences in transcriptional parameters between cells.

In each cell, our model describes the production and degradation of the transcriptional reporter $R$ and consists of two time-dependent and stochastic variables: the transcriptional activity $S$ that acts as a source for the transcriptional reporter $R$ (Fig. 3a). To account for the spread in mean levels (Fig. 2b), $S$ is

allowed to fluctuate around a cell-specific mean, and the variances of $S$ and $R$ are also cell-specific. This flexibility allows the variance of the fluctuations to scale with the mean for each cell, which is expected for gene expression levels.

Reporter levels $R$ are produced at rate $S$ and their effective half-lives were measured independently by blocking transcription with actinomycin D (Supplementary Fig. 5; assumed to be constant across cells for the analysis). This estimated half-life is therefore dependent both on reporter protein and mRNA half-lives. We further introduced the parameter $\rho_{SIS}$ describing the correlation of transcriptional fluctuations between sister cells, which tunes the extent to which sister cells acquire similar reporter profiles over the cell cycle (Fig. 3b). To set the initial conditions, we modelled the mean, variance and co-variance of $R$ and $S$ in the beginning of each cell cycle from the predicted steady-state, assuming $R$ at the beginning of the cell cycle to be at half its steady state value to reflect cell division.

The population model is then built hierarchically, whereby the cell-specific parameters are related to each other through a population level distribution (Fig. 3c), and these population parameters are estimated within our inference scheme. These global parameters therefore control the distribution of cell-specific parameters over all pairs of sister cells, such as the cell-to-cell variability in mean transcription rates. In this model, the population-level parameter $s$ controls the intercellular variability in mean transcription rates (Fig. 3c). The population level can capture the long-tailed distributions typically observed in snap-shot population measurements of gene expression e.g., in smFISH[31]. The correlation in mean transcriptional activity between sister cells is quantified with the parameter $\lambda_{SIS}$ (Fig. 3d), and along with the similarity in dynamics ($\rho_{SIS}$) these two parameters connect sister cells.

Microscopy time traces of single-cell transcriptional reporters are then analysed, and model parameters are estimated within a Bayesian hierarchical framework that combines Gaussian processes with Hamiltonian Markov Chain Monte Carlo (MCMC) sampling for efficient inference (Methods).

**Mean transcriptional activity is correlated between sisters**. To quantitatively understand the finding in Fig. 2, we applied our inference scheme to estimate the parameters of our model for each gene individually (parameter estimates for all genes shown in Supplementary Fig. 6, example trace plots shown in Supplementary Fig. 7).

First, to validate our method, we simulated bioluminescent time series for a range of parameters using 50 pairs of cells and cell-cycle lengths that were similar to our data (Supplementary Note 1), and we found that for all parameters the true values used for simulation were in the 90% credible intervals (Supplementary Fig. 8), showing our method can reliably recover parameters for data that mimicked our experiments.

We next used our model to analyse the gene-specificity of the variability of cell-specific means and correlations between sister cells (Fig. 2). As expected, for individual cells we found that the noise levels for $S$ and $R$ scale with cell-specific mean transcriptional activity (Supplementary Fig. 9). We found that the spread of cell-specific means varies significantly across genes, with *pGK* being the most (coefficient of variation, CV = 0.7) and *Dstn* the least (CV = 0.2) variable (Fig. 4a). This gene-specific variability was only weakly explained by the mean expression level ($R^2 = 0.24$, Supplementary Fig. 10). To test whether cell-specific means are correlated between the two daughters, we analysed the parameter $\lambda_{SIS}$ (Fig. 4b). Interestingly, $\lambda_{SIS}$ was less variable and consistently high across genes, ranging from 0.7 to 0.95 (Fig. 4b). The genes with the highest variability of cell-specific means also

exhibited the highest $\lambda_{SIS}$ (Fig. 4c). Of note this was not due to a structural property of the model, namely, the two parameters were not correlated during inference (Supplementary Fig. 11). This correlation between the spread of cell-specific means and $\lambda_{SIS}$ is qualitatively consistent with a simple model of inheritance in which daughter cells inherit a fraction of the mother's transcriptional activity plus a random component, where the magnitude of this random component is fixed and gene-independent (Supplementary Fig. 12). Below, we investigate the impact of variable cell-specific means on the maintenance of sister-cell correlation over the cell cycle.

**Transcriptional fluctuations show synchronicity in sisters**. We next determined whether the similarity in the dynamics of sister cells we observed (Fig. 2c) could be substantiated by our mathematical model. In the model, similarity of dynamics is quantified with the correlation parameter $\rho_{SIS}$ (ranging from −1 to 1). $\rho_{SIS} = 0$ indicates independent fluctuations in $S$, while $\rho_{SIS} = 1$ indicates identical shapes of transcriptional activity over the cell cycle (for identical initial conditions). Intriguingly, the inferred values of $\rho_{SIS}$ were positive for all genes, confirming that sister cells tend to show correlated dynamics (Fig. 4d). The degree of similarity in dynamics was gene-specific but overall lower than $\lambda_{SIS}$, ranging from $\rho_{SIS} = 0.3$ for *Spry4* to $\rho_{SIS} = 0.7$ for *Jam2*. Having found that correlated transcriptional fluctuations are detectable for all genes, we wanted to further explore the origins of this similarity in dynamics by analysing pairings of randomised non-sister cells (examples shown in Supplementary Fig. 13). If there was cell-cycle dependent transcriptional control affecting all cells, this would lead to a non-zero $\rho$ value even amongst random pairings of cells. In fact, we found that most $\rho_{RAND}$ values were only slightly above zero for random cell pairings, which suggests a modest contribution of cell cycle progression to $\rho$ (Fig. 4e).

The origin of correlated dynamics between sister cells remains unsolved, but one possible explanation is that sister cells share a common microenvironment, and that transcriptional activity could be regulated by local signalling. To address this question, we compared to a control situation in which non-sister cells separated by same average distance as true sister cells were paired. This showed that while $\rho$ was higher for sisters than non-sisters for both *Rbpj* and *Jam2* (Fig. 4f), the value of $\rho$ for non-sisters pairs was still higher than for fully randomised pairings of cells, the latter being on average more spatially distant (compare Fig. 4e, f). Therefore, the microenvironment can, at least in some cases, increase the synchrony in transcriptional dynamics of cells that are close in space.

**Decomposing sister-cell correlations**. We next aimed to investigate how the correlated levels of mean transcriptional activity ($\lambda_{SIS}$) and similarity in dynamics ($\rho_{SIS}$) between sister cells impact the observed loss of correlation between sister cells for each gene over the cell cycle (Fig. 2d). We therefore used the model to predict how the correlation between sister cells evolves over time, using the inferred parameter values (the posterior means) of each gene and the empirical correlation between sisters at the beginning of the cell cycle. Remarkably, comparing the correlation over the cell cycle from the model (red, Fig. 4g–i) with the empirical correlation from the data (green, Fig. 4g–i, all genes shown in Supplementary Fig. 14) showed very good agreement, even if the model was fitted to the time series and hence not directly fitted to this correlation decay. Next, to quantify the relative contributions of different processes in maintaining similar transcriptional levels between sisters, we dropped certain features from the model. First, we set the parameter $s$ to zero (such that all cells share the same mean transcriptional activity), which made the predicted

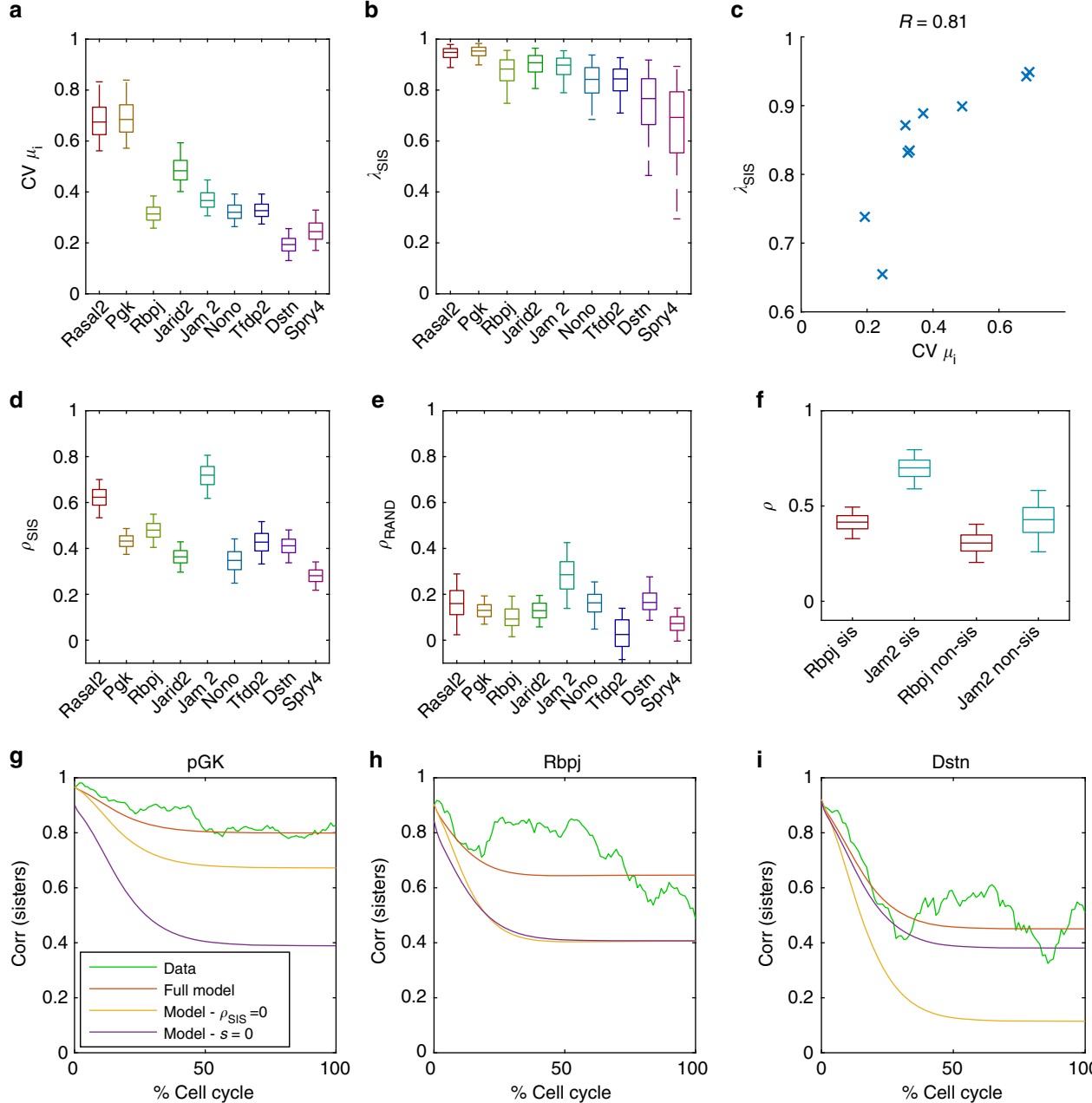

**Fig. 4** Quantification of features contributing to sister-cell correlation. **a** Posterior distributions (shown as boxplots) of the coefficient of variation (CV) for cell-specific means $\mu_i$, calculated from the posterior distributions of $s$ and $m$. The boxplots represent the 25th, median (50th) and 75th percentiles of the posterior distribution and the whiskers represent the 5th and 95th percentiles. **b** Posterior distributions of the correlation of mean transcriptional activity between sister cells ($\lambda_{SIS}$). **c** $\lambda_{SIS}$ correlates with CV of cell-specific means (crosses denote mean posterior values for each gene). **d** The inferred posterior probability distribution of the similarity in dynamics ($\rho_{SIS}$) between sister cells. **e** The inferred posterior probability distribution of the similarity in dynamics ($\rho_{RAND}$) between randomised cells, where the randomisation ensures that cells have the same correlation in cell-cycle lengths as sister cells. **f** The inferred posterior probability distribution of the similarity in dynamics $\rho$ for both sister cells and non-sister cells with the same average distance as non-sister cells. **g–i** Decrease in correlation between sister cells over the cell cycle. Green—the evolution of the sister-cell correlation over the cell cycle from the data, where time is expressed in % of cell cycle time. Red—the parameter posterior means for each gene are used to predict the evolution of sister-sister correlation over the cell cycle from the model, which is normalised to the average cell cycle length (13.5 h). Yellow—the correlation between sisters is recalculated with $\rho_{SIS} = 0$. Violet—the correlation between sisters is recalculated with $s = 0$, which removes cell-specific means from the model

correlation between sister cells decay much faster for most genes (violet, Fig. 4g–i). Similarly, setting $\rho_{SIS}$ to zero led to a faster decorrelation between sister cells (yellow, Fig. 4g–i) (when $\rho_{SIS}$ and $s$ are removed from the model before fitting to data the correlation remains underestimated, showing that both features are required to account for the sister-cell correlation in the data (Supplementary Fig. 15)). Therefore, both $s$ and $\rho_{SIS}$ positively

contributed to the correlation between sister cells, but the relative contributions of these two parameters was gene-specific. For *pGK*, the predicted sister–sister correlation was much lower when variability in cell-specific means was removed ($s = 0$, Fig. 4g), which suggests that variable cell-specific means are important to maintain similar transcriptional activity between sisters for this gene. In contrast, the predicted correlation was not changed

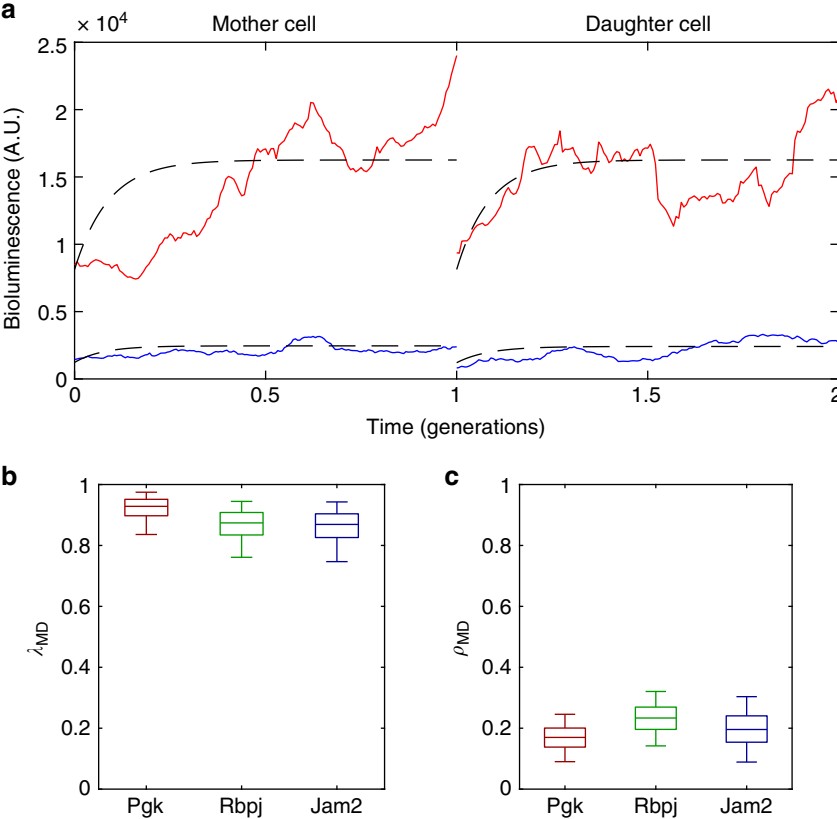

**Fig. 5** Highly correlated transcriptional activities between mother and daughter cells. **a** Examples of two pairs of mother and daughter cells from the *pGK* gene. Red and blue represent different pairs of cells. **b** Posterior distribution (shown as boxplots) of the correlation of mean transcriptional levels ($\lambda_{MD}$) between mother and daughter cells. The box represents the 25th, median (50th) and 75th percentiles of the posterior distributions and the whiskers represent the 5th and 95th percentiles. **c** Inferred posterior probability distribution of the similarity in dynamics ($\rho_{MD}$) between mother and daughter cells

significantly for *Dstn* when variable cell-specific means were abolished ($s = 0$) from the model, and the similarity in dynamics ($\rho_{SIS}$) were more important for maintaining correlation between sisters (Fig. 4i). Our model therefore shows that not only is the maintenance of sister–sister correlation gene-specific, but also that different processes tune sister–sister correlation for different genes.

**Mean transcriptional activity is transmitted to daughters.** Having observed that transcriptional activity is highly correlated between sister cells, we next explored whether transcriptional states were also propagated through cell division. Given that sister cells inherit highly correlated mean transcriptional activities and display similarity in their transcriptional dynamics (Fig. 4b, d), we asked whether this was also the case for mother–daughter pairs. We thus measured the reporter levels of mother and daughter cells for three genes (*pGK*, *Jam2* and *Rbpj*) and re-fitted our model (examples of two pairs shown in Fig. 5a, *pGK* gene). Similarly to the sister cells, cell-specific means between mother and daughter cell were again highly correlated ($\lambda_{MD}$), showing that mean transcriptional activity can be robustly transmitted across generations (Fig. 5b). In contrast, the similarity in dynamics between mother and daughter cells ($\rho_{MD}$) was low (Fig. 5c). Taken together, this data suggests that while the mother cell may to some extent set temporal patterns of transcriptional fluctuations in daughter cells, the shape of fluctuations is largely independent between cell generations. Therefore, the transmission of cell-specific mean transcriptional activity through cell division is the main contributor to the propagation of transcriptional levels from mother to daughter cells. The observed correlation ($\lambda_{MD}$) in mean transcriptional levels between mother and daughter cells for *pGK*, *Jam2* and *Rbpj* were 0.92, 0.87 and 0.86, respectively, suggesting that multi-generational transcriptional memory for those genes persists for 17, 9 and 8 cell generations (correlation times, see Methods).

**Long-term memory of transcriptional levels in cell families.** We next investigated how such transcriptional memory shapes expression levels across families of cells after few (2–3) generations. In such a setting, the analysis of mother–daughter cells implies that inter-family variability would be gene-specific with longer memory genes showing greater spread.

We first measured transcriptional activities within and across families of at least four cells for the *pGK*, *Jam2* and *Dstn* genes (Fig. 6a–c). To minimise biases linked to cell-cycle-related changes in expression levels, we averaged luminescence levels from three image frames preceding nuclear envelope breakdown. We then quantified the inter-family and intra-family variability using a Bayesian hierarchical model where individual cells within each family are distributed around a family mean with an intra-family noise parameter (Methods).

Using model comparison, this clearly showed that there is gene-specificity for intra-family and inter-family transcriptional variability (99.9% weight using WAIC, Methods) (Fig. 6d). Furthermore, the latter was lowest for *Dstn* and highest for *pGK* (Fig. 6d), which was consistent with our sister-cell analysis (Fig. 4a). The differences in intra-family variability were overall small between genes (Fig. 6d). This is not surprising since we expected intra-family variability to scale with the magnitude of relative reporter fluctuations (i.e., the standard deviations of

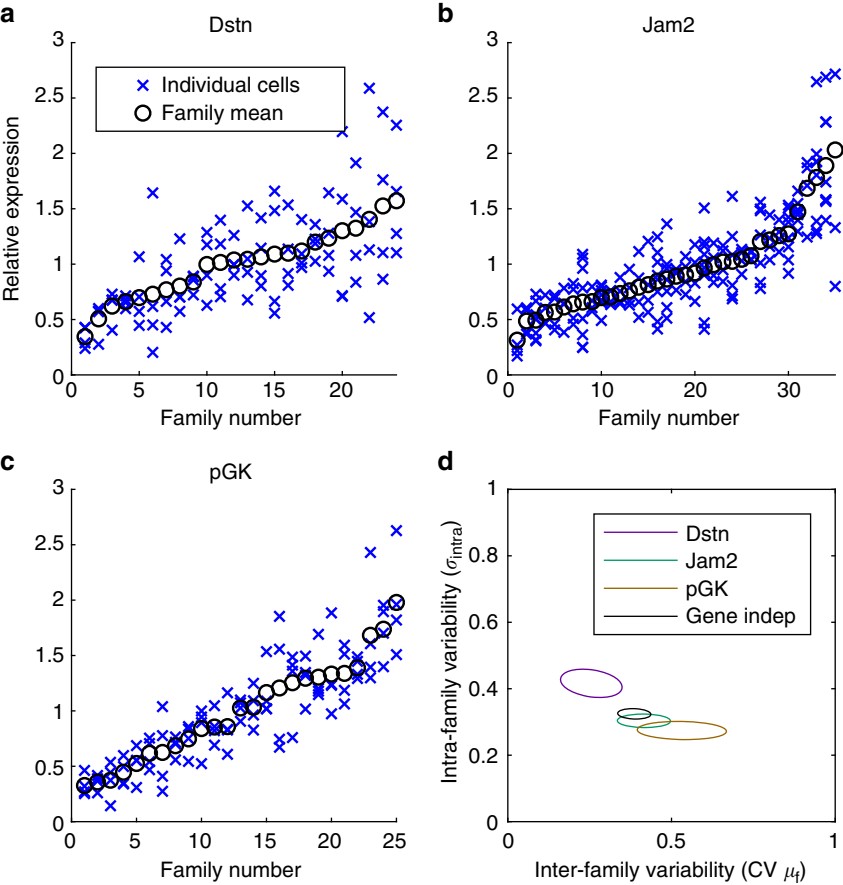

**Fig. 6** Spread and multi-generational memory of transcriptional levels. **a–c** Distribution of transcriptional reporter levels in different families. For each family, crosses represent averaged luminescence levels from three frames preceding nuclear envelope breakdown in individual cells. Circles represent the mean of the family. **d** Posterior distributions of the parameters from the Bayesian hierarchical model for family data (see Methods). The inter-family variability represents the CV of family means ($\mu_f$), which is calculated from $m$ and $s$, and the intra-family variability is represented with $\sigma_{intra}^2$. Ellipses represent posterior means ± SD. The black ellipse represents the gene-independent model, where the parameters of the model were assumed to be the same for all three genes

reporter fluctuations/mean), which was quite similar across three different genes ($Dstn = 0.39$, $Jam2 = 0.31$, $pGK = 0.37$). Overall, gene-specific differences in transcriptional memory thus create characteristic distributions of average transcription levels across families of related cells, where the inter-family spread of transcription is larger for long-memory genes.

## Discussion

One of the major challenges in quantitative biology is to understand how gene expression dynamics of single cells are related in the context of multicellularity. The combination of lineage-tracing and mRNA measurements has previously been used to quantify the dynamics of cell-fate transitions[32,33]. However, thus far, it remained unclear how transcriptional fluctuations are propagated in lineages of phenotypically homogenous cells, and to which extent this transmission is gene-specific. Previous studies in fixed mammalian cell lines have reported higher similarity of mRNA levels of neighbouring cells[11] and that population context can predict cellular features such as membrane lipid composition and endocytosis[15], but the impact of lineage relationships on such microenvironment-related correlations was not addressed in these studies.

Lineage information was found to be an important contributor to patterning gene expression in bacterial microcolony formation[34,35], where it can act as the dominant cause of spatial correlations[36]. While properties such as cell cycle duration have

been shown to propagate in mammalian cell lineages[37,38], the importance of genealogy for transcriptional activity in mammalian cells is still poorly studied. Here, we used live-cell imaging to measure and compare transcriptional activity of lineage-related mammalian cells over time. We developed a simple yet powerful stochastic model of gene expression fluctuations, which combined with Bayesian inference allowed us to identify the key processes and parameters underlying the observed correlation patterns of transcriptional reporter levels within lineage-related cells. This quantitative analysis allowed us to separate short-term transcriptional fluctuations from long-term trends, which both contribute to population heterogeneity in the dynamics of the observed reporter levels.

In particular, we found that transcriptional activities in each cell within the population fluctuate around cell-specific mean levels, which propagate through cell division in a gene-specific manner and result in multi-generational transcriptional memory. The high correlation in transcriptional activity between mother and daughter cells implied a memory time-scale of up to 17 generations. Certain properties such as the cell cycle durations show complex patterns of inheritance whereby there is a higher correlation between mother and granddaughter cells than between mother and daughter cells[38]. Future studies might show if transcriptional activities follow similar inheritance patterns. Remarkably, the rate at which transcriptional activity of sister cells diverge from each other was correlated with the spread of

transcriptional activity in the population (Fig. 4c). This suggests that the duration of transcriptional memory scales with the range of expression in the population, while the relative change in transcriptional activity per generation appears to be more conserved across genes.

Surprisingly, sister cells displayed not only similar cell-specific mean levels, but also correlation of their transcriptional dynamics over the cell cycle. Our model uses a dedicated parameter ($\rho$) to capture similarity in dynamics, which contrasts with previous mathematical modelling assuming that transcriptional dynamics of sister cells are independent[39,40]. At the mechanistic level, this correlation in dynamics could be caused by correlated inheritance of factors from the mother cell that control transcriptional dynamics. The much lower correlated dynamics between mother and daughter cell pairs suggest that the factors controlling transcriptional activity may change significantly over the course of one cell cycle, and thus set a different transcriptional program in the next cell generation. For some of the genes, non-sister cells in the same spatial proximity as sister cells also exhibited correlated transcriptional fluctuations (Fig. 4f). While we cannot exclude that such non-sister cells could still be related (e.g., cousins), the correlated transcriptional fluctuations could also be due to spatially proximal cells being exposed to similar microenvironments. Such similarity might involve shared extracellular signals or number of neighbouring cells[15]. For example, in *Dictyostelium*, spatial clustering in the timing of transcriptional bursts was linked to local signalling[41]. The microenvironment could also explain the higher similarity in dynamics between pairs of sister cells compared to mother–daughter pairs. Of note, inherited and microenvironmental factors may have indistinguishable consequences on transcriptional dynamics similarity of proliferating adherent cells, since related cells will typically remain in close spatial proximity.

Several potential regulators could determine the timescale of transcriptional memory, such as mitotic bookmarking transcription factors, histone modifications, DNA methylation, or spatial DNA positioning[1,42,43]. While physiological parameters such as cell size variability could explain differences in mRNA counts across cells[11], these global factors are unlikely to fully explain our data as they are common for all genes examined, and for example the CV of the cell-specific transcriptional activities ranges from 0.2 to 0.7 across the genes we measured (Fig. 4a). Whether potential gene-specific regulators of memory involve *cis*-regulatory or *trans*-regulatory factors will be the subject of future studies.

The findings we describe here suggest a potential role for transcriptional memory in tissue patterning during developmental processes. This passive mechanism could thereby be the prime changes in expression patterns between groups and families of cells, which may be further reinforced and stabilised by diverging cell fate decisions.

## Methods

**Generation of lentiviral constructs**. To generate the pSTAR-GTX gene trap lentiviral vector, ten repeats of the 9-nucleotide IRES element derived from the 5′ UTR sequence of the *gtx* mRNA (Chappell, Edelman, Maura, PNAS 2000), interspersed with 9-nt spacers based on a segment of the β-globin 5′ UTR (nt 9–17), were inserted upstream of bsdF2ANLSLuc by restriction cloning into the pSTAR lentiviral vector[44]. To generate the pGK-Luc lentiviral construct, the pGK promoter was PCR-amplified from the pLV-pGK-rtTA3G-IRES-Bsd[44] and inserted upstream of bsdF2ANLSLuc by restriction cloning into the pSTAR lentiviral vector.

**Stable cell line generation**. The stable gene trap (GT) cell lines were generated by transducing E14 mouse embryonic stem (ES) cells (kindly provided by Didier Trono, EPFL) with the concentrated virus carrying the pSTAR-GTX or pGK-Luc construct. Virus production was performed by co-transfection of HEK 293T cells with the construct of interest, the envelope (PAX2) and packaging (MD2G)

constructs using calcium phosphate, and concentrated 120-fold by ultra-centrifugation as described previously[20]. ES cells were then seeded at a density of 125,000 cells per 10 cm dish and transduced with 125 μl of virus. Antibiotic selection was started by addition of 10 μg/ml of blasticidin 3 days after transduction, while the outgrown colonies were picked 14–21 days after. The small number of outgrown colonies per 10 cm dish (two on average) ensured we obtained a single active insertion per clone. Colonies were then expanded in the selection medium and subsequently frozen. The FUCCI ES cell line was generated by transducing ES cells with 50 μl of 120-fold concentrated lentiviral vectors encoding mKO2-hCdt1 and mAG-hGem[45], followed by FACS to sort cells positive for both mKO2 and mAG fluorescence.

**Cell culture**. All ES cell lines were derived from the E14 cell line (kind gift from Didier Trono, EPFL), cultured at 37 °C and 5% $CO_2$, on dishes coated with 0.1% gelatin type B (Sigma), in GMEM (Sigma) medium supplemented with 10% ES cell-qualified FBS, 1× nonessential amino acids (NEAA), 2 mM L-glutamine, sodium pyruvate, 100 μM 2-mercaptoethanol, 1% penicillin and streptomycin, home-made leukaemia inhibitory factor (LIF), CHIR99021 at 3 μM and PD184352 at 0.8 μM. Cells were split every 2–3 days. The pGK-Luc cell line was constantly maintained in the presence of 10 μg/ml of blasticidin to prevent silencing of the reporter.

HEK 293T (ATCC) cells were cultured at 37 °C and 5% $CO_2$, in DMEM medium (Sigma) supplemented with 10% FBS and 1% penicillin and streptomycin (BioConcept, 4-01F00H).

**Mapping of insertion sites in gene trap cell lines**. To identify the endogenous gene into which the pSTAR-GTX was inserted in each GT cell line, we used splinkerette PCR (spPCR)[28] with modified primer sequences adapted to our lentiviral gene trap construct (Supplementary Table 1). This method allows the amplification of a portion of DNA between the GT cassette and a known DNA sequence (adaptor). Genomic DNA (gDNA) was extracted from cells of each clone using the Qiagen gDNA Extraction Kit (Qiagen). gDNA was cut with 4-cutter restriction enzyme MluCI, followed by ligation to the annealed small and long adaptor. The ligation was followed by HindIII digestion, allowing removal of the adaptors and most of the GT cassette. Then, the portion of DNA between the adaptor and the GT cassette was amplified through two rounds of PCR. The bands from the nested PCR were purified using the QIAquick gel extraction kit (Qiagen) and directly sequenced using nested primers (Supplementary Table 2; F2 and R2). Sequences derived from spPCR were used to identify the insertion site through the BLAT genome alignment tool (http://genome.ucsc.edu) (Supplementary Fig. 2)[46]. At the same time, since MluCI and EcoR V cut both LTRs, an additional 200 bp DNA segment was amplified in all samples, which was used as a control of successful nested PCR amplification.

**Luminescence microscopy**. Luminescence imaging was performed on an Olympus LuminoView LV200 microscope equipped with an EM-CCD camera (Hamamatsu photonics, EM-CCD C9100-13), a 60-fold oil-immersion magnification objective (Olympus UPlanSApo 60x, NA 1.35, oil immersion) in controlled environment conditions (37 °C, 5% $CO_2$). Sixteen to 24 h before imaging, 50,000–75,000 cells were seeded on FluoroDishes (WPI, FD35-100) coated with E-cadherin, allowing to obtain a monolayer of individual cells suitable for single cell tracking[47]. The medium was supplemented with 0.5 mM luciferin (NanoLight Technology, Cat#306 A) two to four hours before imaging. Fields of view with about 10 to 30 cells were imaged every 5 min with an exposure time of 299 s for 24 to 48 h. To examine propagation of gene expression levels within ES cell colonies (Fig. 6d), 500–1000 cells were seeded on Fluorodishes coated with gelatin, and grown as colonies for 60 h. For each clone, two consecutive images with an exposure time of 5 (Dstn and Jam2) or 3 (pGK) min in at least 10 fields of view were acquired.

**Reporter half-life measurements**. Single cell reporter half-lives were determined by treating cells with 5 μg/ml of Actinomycin D, which inhibits RNA elongation and thus results in transcriptional arrest[48]. Luminescence imaging was performed as described above for 3 to 5 h, starting immediately after addition of Actinomycin D. Although both protein and mRNA half-lives contribute to overall reporter half-life, the decay curve was well fitted by a first order exponential function (Supplementary Fig. 5).

**Cell cycle phase durations**. In order to determine the durations of the different cell cycle phases, we combined different approaches. We first used time-lapse imaging of ES cells expressing both components of the FUCCI system[49] to measure the duration of the whole cell cycle and of G1 phase. The FUCCI systems relies on biphasic cell cycle-dependent activity and proteolysis of the ubiquitination oscillators Cdt1 and Geminin, whose fragments are fused to mKO2 and mAG, respectively. Cells were seeded on E-cadherin at a density of 50,000 cells per well of a black 96-well plate (Sigma) 16 to 24 h before imaging. Time-lapse fluorescence imaging was performed using an inverted Olympus Cell xCeed with a ×20 objective (Olympus UPlanSApo 20x, NA 0.75) in controlled environment conditions (37 °C, 5% $CO_2$). Green and red fluorescence were measured using the GFP and Cy3

channel, respectively, every 10 min with an exposure time of 300 ms for 24 h. The fluorescence time-lapse acquisitions were analysed manually using the Fiji software. mKO2 expression allowed us to define the duration of G1, while mAG was expressed in the S, G2 and M phases. To directly measure the length of M phase in mES cells, we used single cell traces from the luminescence time-lapse acquisitions in which nuclear envelope breakdown is clearly visible as a sudden increase in the area occupied by the luminescence signal of an individual cell. We thus manually determined the number of frames from the moment of nuclear breakdown in the prophase of the cell cycle, until the moment when we see formation of two new nuclei manually. Using this information we were able to calculate the average length of M phase in single cells (Supplementary Fig. 16).

**Cell tracking and image analysis**. Prior to quantification of single cell gene expression from luminescence microscopy movies, we removed imaging artifacts known as cosmic rays using the Min operation of the Fiji software Image Calculator function. To track cells, we used Fiji to manually draw the outlines around cells, using a fixed area with shape adjustment when required. Background measurements were performed close to every tracked cell, in regions devoid of luminescent signal separately for each time point of the movie, and these values were subtracted from cell measurements. Cells were tracked from the time they were born (just after division of their mother cell) until the last frame before cytokinesis, either as pairs of sisters or pairs of mother and daughter cells. For the experiments investigating the impact of microenvironment on similarity of gene expression between cells, the distance between sister cells and non-sister cells was measured by hand-drawing a line between the approximate centres of two nuclei. The distance between sister cells was measured every ten frames for 500 min, starting from the tenth frame after their birth. In the case of non-sister cells, the distances between cells present over the same time period in the field of view were measured every ten frames for 500 min. For the *Jam2* gene, the average distance of sisters was $0.07 \pm 0.01$ µm (standard error), and for non-sisters the average distance was $0.08 \pm 0.01$ µm. For the *Rbpj* gene, the average distance of sisters was $0.11 \pm 0.01$ µm, and for non-sisters the average distance was $0.12 \pm 0.01$ µm. We defined the first measurement as the time frame when the later cell in a pair was born. Additionally, for the cell family experiments (Fig. 6a–c), we tracked families of 4 cells that were from the middle towards the end of the cell cycle for 3 frames.

**Single-cell reporter level dynamics**. The objective of the mathematical model was to capture the key processes that underlie the observed correlation patterns of transcriptional reporter levels within lineage-related cells. We first describe the stochastic model of single-cell dynamics that captures noisy fluctuations amongst pairs of cells, and then describe how cell-specific parameters are connected via a population model. Parameter inference of the model is performed for each gene using Markov Chain Monto Carlo within a Bayesian framework.

For two sister cells labelled i ∈ {1, 2}, we model the total production rate of the bioluminescent reporter with the variable $S$, which we interpret as a total transcriptional activity. The dynamics of the transcriptional activity for cell $i$ follows the stochastic differential equation

$$\frac{dS_i(t)}{dt} = -\frac{1}{\tau_S}\big(S_i(t) - \mu_i\big) + \epsilon_i(t), \tag{1}$$

where the first term describes the relaxation to a cell-specific mean level ($\mu_i$). The time scale $\tau_S$ controls the rate at which $S$ fluctuates (i.e., slow or rapid fluctuations for large or small $\tau_S$, respectively). The distribution of the cell-specific means $\mu_i$ is further modelled at the population level (described below). The term $\epsilon_i(t)$ models biological noise, for example arising from the stochastic biochemical processes occurring in single cells, and acts to continuously deliver random perturbations to the transcriptional activity. $\epsilon_i(t)$ is modelled as Gaussian white noise with zero mean and variance

$$\text{Cov}[\epsilon_i(t)\epsilon_i(t')] = \frac{2\sigma_{S,i}^2}{\tau_S}\delta(t - t'), \tag{2}$$

where $\sigma_{S,i}^2$ controls the size of the perturbations on $S_i$ and is cell specific. In the stationary state, the covariance of the transcriptional activity is $\text{Cov}[S_i(t)S_i(t')] = \sigma_{S,i}^2$. To account for the similarity in dynamics observed in sister cells we introduced a correlation parameter $\rho_{SIS}$ linking the noise terms of two sisters:

$$\text{Cov}[\epsilon_1(t)\epsilon_2(t')] = \rho_{SIS}\frac{\sigma_{S,1}\sigma_{S,2}}{\tau_S}\delta(t - t'). \tag{3}$$

$\rho_{SIS}$ can vary between −1 and 1. When $\rho_{SIS} = 0$ the cells are fluctuating independently and have uncorrelated trajectories, but when $\rho_{SIS} > 0$ (or $\rho_{SIS} < 0$) the perturbations are correlated (or anti-correlated) between cells.

The measured total transcriptional reporter level is modelled with the variable $R$. The reporter $R$ is produced at rate $S$ and is degraded with half-life $\tau_R$

$$\frac{dR_i(t)}{dt} = \frac{\ln(2)}{\tau_R}\big(S_i(t) - R_i(t)\big) + \eta_i(t), \tag{4}$$

where $\eta_i(t)$ corresponds to noise at the reporter level. Note that to save parameters, mRNA is not explicitly modelled; we estimated the net reporter half-life (which

thus depends on both the mRNA and protein half-life) by blocking transcription with actinomycin D and by fitting a first order exponential decay to the decrease in reporter levels (values shown in Supplementary Fig. 5). $\eta_i(t)$ is taken as Gaussian white noise, and represents effective noise combining both molecular fluctuations in reporter levels, as well as experimental noise. $\eta_i(t)$ is assumed to be independent between two cells. The variance of the reporter Gaussian white noise terms is given by

$$\text{Cov}[\eta_i(t)\eta_i(t')] = \frac{2\sigma_{R,i}^2\ln(2)}{\tau_R}\delta(t - t'), \tag{5}$$

where $\sigma_{R,i}^2$ controls the cell-specific variance in reporter levels. Our model consists of a system of two linear stochastic differential equations (Eqs. 1 and 4), and if the initial conditions of the two variables are normally distributed then the model can thus be analysed within the framework of Gaussian processes (Supplementary Note 1). Note that while the time series of individual cells are modelled as Gaussian processes, the presence of cell-specific parameters and a population model means that the total distribution over all cells is more complex than a simple Gaussian distribution (i.e., it is a mixture of Gaussians with different means and variances).

**Initial conditions**. As for any system of SDEs the distribution for the initial conditions at time $t = 0$ (i.e., following cell division) need to be specified. Here, the distributions over $R$ and $S$ were taken from the steady state solution of the model, with the modification that the $R$ variable was divided by two, reflecting the fact that we measure the total levels of transcriptional reporter, which are approximately halved at cell division (Supplementary Note 1).

**Population level**. The above model (Eqs. 1 and 4) introduced cell-specific mean levels $\mu_i$ (Fig. 2b), as well as cell-specific transcriptional noise ($\epsilon_i$) and noise in reporter dynamics ($\eta_i$). Across the population, we assumed that these quantities are log-normally distributed. For example, this captures the heavy tails of expression levels (e.g., data in Fig. 2a show few high-expressing cells). Moreover, we introduce a parameter $\lambda_{SIS}$ representing the correlation in mean transcriptional activities between sister cells (i.e., the population correlation between $\mu_1$ and $\mu_2$ for pairs of cells across the population). Together the population distributions of $\mu_i$, $\sigma_{S,i}$ and $\sigma_{R,i}$ are parameterised as follows

$$\log\begin{pmatrix}\mu_1\\\mu_2\end{pmatrix} \sim N\left(\begin{bmatrix}m\\m\end{bmatrix}, \begin{bmatrix}s^2 & \lambda_{SIS}s^2\\\lambda_{SIS}s^2 & s^2\end{bmatrix}\right), \tag{6}$$

$$\log\begin{pmatrix}\sigma_{S,1}\\\sigma_{S,2}\end{pmatrix} \sim N\left(\begin{bmatrix}\Lambda_S\\\Lambda_S\end{bmatrix}, \begin{bmatrix}\Sigma_S & 0\\0 & \Sigma_S\end{bmatrix}\right), \tag{7}$$

$$\log\begin{pmatrix}\sigma_{R,1}\\\sigma_{R,2}\end{pmatrix} \sim N\left(\begin{bmatrix}\Lambda_R\\\Lambda_R\end{bmatrix}, \begin{bmatrix}\Sigma_R & 0\\0 & \Sigma_R\end{bmatrix}\right). \tag{8}$$

Where $N$ stands for a 2-variable normal distribution. Thus, the population mean of log $\mu_i$, $\sigma_{S,i}$ and $\sigma_{R,i}$ are parameterised with $m$, $\Lambda_S$ and $\Lambda_R$, respectively. The intercellular population variances of $\mu_i$, $\sigma_{S,i}$ and $\sigma_{R,i}$ are parameterised with $s^2$, $\Sigma_S$ and $\Sigma_R$, respectively. When $\lambda_{MD}$ represents the correlation in mean transcriptional activities between mother and daughter cells, we calculate a correlation time as the number of generations it would take for the value of $\lambda_{MD}$ to decay to 1/e.

**Parameter inference**. Because of the population parameters the full model is a so-called hierarchical model. Parameter inference for each gene was performed within a Bayesian framework. The joint posterior distribution over all parameters (of all cell pairs of a given gene) was inferred using Hamiltonian Markov Chain Monte Carlo (MCMC) sampling, which uses the gradients of the posterior to improve the efficiency of the sampling. We discarded the first 200 samples of each chain as burn-in and then obtained 2500 samples from 4 parallel chains. The inference procedure (including the priors for all parameters) is fully described in Supplementary Note 1.

**Statistical analysis of family data**. For analysing the family data (of at least 4 cells) we used a Bayesian hierarchical model. Each family $f$ was assumed to have a mean transcriptional activity $\mu_f$ and the distribution was modelled with a log-normal distribution.

$$\log\big(\mu_f\big) \sim N\big(m, s^2\big). \tag{9}$$

For the intra-family variability, the reporter levels $R_i$ of each family member $i$ in family $f$ was then modelled as being normally distribution around this family mean

$$R_i \sim N\big(\mu_f, \sigma_{\text{intra}}^2\big), \tag{10}$$

where the variance of this distribution $\sigma_{\text{intra}}^2$ effectively contains contributions from both $S$ and $R$ from the previous model, but which cannot be separated in this data as it provides only a static snapshot. We fit this model within pyStan using 4 chains of 5000 samples[50].

We proposed two different models and found the statistical weight for each of them using widely-applicable information criterion (WAIC) and Akaike weights[51]. In model 1, we assumed that the family data distributions were gene-dependent, and hence each gene had its own set of parameters. In model 2, we assumed a common set of parameters for all three genes considered. We estimated the WAIC using samples from the inferred posterior of the model, which are labelled $\theta^s$, $s = 1$, …, $S$:

$$\text{WAIC} = -2 \sum_{n=1}^{N} \log\left(\frac{1}{S} \sum_{s=1}^{S} p(R_n | \theta^s)\right) + 2 \sum_{n=1}^{N} V_{s=1}^{S}(\log(\text{p}(R_n | \theta^s))), \tag{11}$$

where $V_{s=1}^{S}$ is the sample variance and $N$ represents the total number of cells in the experiment. For model comparison we calculated the Akaike weight $w_{\text{dep}}$ for the gene-dependent model using

$$w_{\text{dep}} = \frac{\exp\left(-\frac{1}{2}\text{dWAIC}_{\text{dep}}\right)}{\exp\left(-\frac{1}{2}\text{dWAIC}_{\text{dep}}\right) + \exp\left(-\frac{1}{2}\text{dWAIC}_{\text{ind}}\right)}, \tag{12}$$

where dWAIC is the difference between each WAIC and the lowest WAIC.

**Reporting summary**. Further information on experimental design is available in the Nature Research Reporting Summary linked to this article.

## Data availability

Data generated during the study and code to generate all figures are available at https://github.com/Nick-E-P/TranscriptionalMemory. The data that support the findings of this study are available from the corresponding author upon request.

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

## Acknowledgements

This work in the Naef lab was supported by the EPFL. Work in the Suter lab was supported by the Swiss National Science Foundation (grant #PP00P3_144828).

## Author contributions

A.M., D.M.S. and F.N. conceived the study. A.M., N.E.P., S.O., F.N. and D.M.S. developed the methodology. N.E.P and F.N. performed the analysis and developed the code. N.E.P., D.M.S. and F.N. wrote and edited the manuscript. A.M. performed experiments and edited the manuscript. D.M.S. and F.N. acquired funding and provided both resources and supervision. D.M.S. and F.N. contributed equally to this work.

## Additional information

**Competing interests:** The authors declare no competing interests.

**Journal Peer Review Information**: *Nature Communications* thanks the anonymous reviewers for their contribution to the peer review of this work. Peer reviewer reports are available.

