## [Peer Review File · Nature Communications]

Reviewers' comments:

Reviewer #1 (Remarks to the Author):

This manuscript by Suter monitors gene expression, using a short-lived luminescence reporter, in mouse ES cells over 1-2 generations, and identifies sister-sister correlations and mother-daughter correlations, implying, by percolation of the luminescence data through a mathematical model, some kind of transcriptional memory. The work also suggests some contribution to these correlations from the micro-environment, in addition to some input from shared cell cycle state. Another feature of the data is that genes with expression showing a greater variance tend to have bigger correlations in related cells.

The work seems sound, and is well presented. The quantitative aspects are clearly outlined for a more traditional biologist. The work recalls several earlier studies dealing with the memory issue, such as Sigal et al (who looked at protein fluctuation time) and some more recent studies using direct transcriptional reporters. Although not explicitly dealing with "memory", a number of labs (Timm Schroeder, Elowitz and others) have shown that the pluripotency factor Nanog has long fluctuation timescales- up to 10 cell generations for Schroeder. Even with a protein reporter (Schroeder's was destabilised), this lifetime can not be accounted for by reporter stability, as 10 cell generations is a lot of dilution. So by inference, there is a transcriptional memory here already, albeit for 1 gene.

The novel features of the present work by Suter are the use of multiple genes, which allows observation of gene specific effects (although these were also remarked upon by the Sigal paper), and the trend of the more variable genes having the greater apparent "memory". This feature- that in a population with a high variance, things that start the same will appear to be more correlated- is this simply a truism?

The concept of a transcriptional memory was frequently controversial, especially with biochemists (see the debates centred around Ptashne, more recently the review by Danny Reinberg), so it is important these studies are carried out and reported.

Is it possible some of the correlations between sisters can be accounted for because they are measured at the same time in the culture? Cells dividing later might experience different culture effects, and so randomising them (to generate some kind of expected null correlation value) will impose a time-caused bias.

Reviewer #2 (Remarks to the Author):

The manuscript of Philips and Mandic et al. is an interesting read containing nice observations and describing useful experimental and analytical tools. Though the lack of more refined mechanistic insight is a bit dissatisfying at occasions, the necessary studies would largely seem outside of the scope of this paper, and I hence hope that the publication of this manuscript will encourage such investigation. With some rare exceptions, the authors are very self-critical about possible interpretations of their observations and tools and make these nuances transparent to readers of distinct fields.

Reviewers' comments:

Reviewer #1 (Remarks to the Author):

This manuscript by Suter monitors gene expression, using a short-lived luminescence reporter, in mouse ES cells over 1-2 generations, and identifies sister-sister correlations and mother-daughter correlations, implying, by percolation of the luminescence data through a mathematical model, some kind of transcriptional memory. The work also suggests some contribution to these correlations from the micro-environment, in addition to some input from shared cell cycle state. Another feature of the data is that genes with expression showing a greater variance tend to have bigger correlations in related cells.

The work seems sound, and is well presented. The quantitative aspects are clearly outlined for a more traditional biologist. The work recalls several earlier studies dealing with the memory issue, such as Sigal et al (who looked at protein fluctuation time) and some more recent studies using direct transcriptional reporters. Although not explicitly dealing with “memory”, a number of labs (Timm Schroeder, Elowitz and others) have shown that the pluripotency factor Nanog has long fluctuation timescales- up to 10 cell generations for Schroeder. Even with a protein reporter (Schroeder’s was destabilised), this lifetime can not be accounted for by reporter stability, as 10 cell generations is a lot of dilution. So by inference, there is a transcriptional memory here already, albeit for 1 gene.

We thank the reviewer for the positive comments and we’re glad that the paper is accessible for traditional biologists. However, the alluded similarity to previous work is misleading. Nanog fluctuations reflect the transition between different cell types, namely naive embryonic stem cells and primed (or epiblast) stem cells. Therefore, these are specifically investigating phenotypical memory and not transcriptional memory within a phenotypically stable cell population, as we do. Concerning the work by Alon and coworkers (Sigal et al., Nature 2006), these investigate protein memory, which heavily depends on mRNA and protein half-lives hence preventing from a direct analysis of transcriptional memory.

In contrast to these earlier studies, here we focussed on transcriptional memory within a population of phenotypically homogenous cells. To better illustrate this key distinction, we have referenced the work from the Schroeder lab in the Introduction (page 3),:

“For instance, in mouse embryonic stem cells (mESCs) exhibiting reversible phenotypic transitions between naïve and primed states, it was found that transitions between different NANOG protein levels can exceed one generation, and after sorting for low NANOG levels there is a subpopulation without NANOG onset for 70 hours, presumably as a consequence of these transitions¹⁸”.

We have also substantially re-written the introduction to clearly define what we mean by transcriptional memory.

The novel features of the present work by Suter are the use of multiple genes, which allows observation of gene specific effects (although these were also remarked upon by the Sigal paper), and the trend of the more variable genes having the greater apparent “memory”. This feature- that in a population with a high variance, things that start the same will appear to be more correlated- is this simply a truism?

Definitely not. In order to better emphasize the novelty of our findings, we have added Supplementary Figure 12. In the previous Figure 4c we demonstrated a strong relationship between the spread of mean transcriptional activity in the population and the correlation in transcriptional activity between sister cells.

Our new Supplementary Fig. 12 shows how this result can be interpreted using a simple model of how transcriptional activity is transmitted from mother to daughter cells. In this model, the daughter cells inherit a fraction of the mother’s activity with the addition of noise.

There are two intuitive interpretations of how transcriptional activity propagates in this model. In scenario 1, the inherited fraction of mother’s activity is the same but the strength of the noise varies for different genes. In this scenario, there is no correlation between the spread of the transcriptional activity (CV_{μ_i}) and the correlation between sisters λ_{SIS} (Supplementary Fig. 12a), hence showing that this is not trivial as the referee suggests. In scenario 2, the inherited fraction of mother’s activity can vary across genes but the strength of noise is fixed. This scenario leads to a correlation between the spread of the transcriptional activity (CV_{μ_i}) and the correlation between sisters λ_{SIS} (Supplementary Fig. 12b).

Therefore, the finding that genes with a large spread of transcriptional activities have higher correlated daughter cells is rather deep, since it indicates that the noise strength which displaces transcriptional activity levels from generation to generation is gene-independent. We have now also added this to the bottom of page 7:

“This correlation between the spread of cell-specific means and λ_{SIS} is qualitatively consistent with a simple model of inheritance in which daughter cells inherit a fraction of the mother’s transcriptional activity plus a random component, where the magnitude of this random component is fixed and gene-independent (Supplementary Fig. 12).”

The concept of a transcriptional memory was frequently controversial, especially with biochemists (see the debates centred around Ptashne, more recently the review by Danny Reinberg), so it is important these studies are carried out and reported.

We thank the reviewer for appreciating the importance of the topic we are focusing on here.

Is it possible some of the correlations between sisters can be accounted for because they are measured at the same time in the culture? Cells dividing later might experience different culture effects, and so randomising them (to generate some kind of expected null correlation value) will impose a time-caused bias.

We thank the reviewer for making this important point, which we in fact already addressed in the submitted version. Indeed, it is possible that sharing a similar environment in time and space may contribute to the similarity between sister cells. We explicitly test this (Figure 4f) and show that non-sister cells with the same average distance as sister cells indeed have higher similarity than randomised pairings (Rbpj, random = 0.10, non-sisters (with same distance as sisters) = 0.31, sisters = 0.41; Jam2 random = 0.29, non-sisters = 0.43, sisters = 0.70). This shows that environmental effects alone are not sufficient to explain the similarity in transcriptional dynamics between sister cells. We have now clarified our conclusions from this result in the Discussion (page 13):

“For some of the genes, non-sister cells in the same spatial proximity as sister cells also exhibited correlated transcriptional fluctuations (Fig. 4f). While we cannot exclude that such non-sister cells could still be related (e.g. cousins), the correlated transcriptional fluctuations could also be due to spatially proximal cells being exposed to similar microenvironments. Such similarity might involve shared extracellular signals or number of neighbouring cells¹⁵. For example, in Dictyostelium, spatial clustering in the timing of transcriptional bursts was linked to local signalling⁴². The microenvironment could also explain the higher similarity in dynamics between pairs of sister cells compared to mother-daughter pairs. Of note, inherited and microenvironmental factors may have indistinguishable consequences on transcriptional dynamics similarity of proliferating adherent cells, since related cells will typically remain in close spatial proximity.”

Reviewer #2 (Remarks to the Author):

The manuscript of Philips and Mandic et al. is an interesting read containing nice observations and describing useful experimental and analytical tools. Though the lack of more refined mechanistic insight is a bit dissatisfying at occasions, the necessary studies would largely seem outside of the scope of this paper, and I hence hope that the publication of this manuscript will encourage such investigation. With some rare exceptions, the authors are very self-critical about possible interpretations of their observations and tools and make these nuances transparent to readers of distinct fields.

We thank the reviewer for her/his positive assessment of our work.

REVIEWERS' COMMENTS:

Reviewer #1 (Remarks to the Author):

The authors have responded effectively to my concerns. The article is now ready for publication.

Reviewer #2 (Remarks to the Author):

I would like to thank the Phillips and colleagues for their revision and recommend the publication of their manuscript.

Concerning the confusion surrounding the usage of Gaussian distributions to describe parts of the variation, I'd like to refer the authors to pages 20 and 21 of their methods section, which they may revise at their discretion.

REVIEWERS' COMMENTS:

Reviewer #1 (Remarks to the Author):

The authors have responded effectively to my concerns. The article is now ready for publication.

We thank the reviewer for her/his appreciation of our work

Reviewer #2 (Remarks to the Author):

I would like to thank the Phillips and colleagues for their revision and recommend the publication of their manuscript.

Concerning the confusion surrounding the usage of Gaussian distributions to describe parts of the variation, I'd like to refer the authors to pages 20 and 21 of their methods section, which they may revise at their discretion.

We thank the reviewer for her/his appreciation of our work. To clarify the use of Gaussian distributions we have now added an additional sentence on page 21:

"Note that while the time series of individual cells are modelled as Gaussian processes, the presence of cell-specific parameters and a population model means that the total distribution over all cells is more complex than a simple Gaussian distribution (*i.e.* it is a mixture of Gaussians with different means and variances)."